# Using Smart Sensors to Monitor Physical Activity and Technical–Tactical Actions in Junior Tennis Players

**DOI:** 10.3390/ijerph17031068

**Published:** 2020-02-07

**Authors:** José María Giménez-Egido, Enrique Ortega, Isidro Verdu-Conesa, Antonio Cejudo, Gema Torres-Luque

**Affiliations:** 1Department of Physical Activity and Sport, University of Murcia, Regional Campus of International Excellence “Campus Mare Nostrum”, Faculty of Sport Science, 30720 Murcia, Spain; josemaria.gimenez@um.es (J.M.G.-E.); antonio.cejudo@um.es (A.C.); 2Department of Languages and Computer Systems, University of Murcia, Regional Campus of International Excellence “Campus Mare Nostrum”, Faculty of Informatics, 30100 Murcia, Spain; iverdu@um.es; 3Faculty of Humanities and Education Sciences, University of Jaen, 23071 Jaen, Spain; gtluque@ujaen.es

**Keywords:** smart sensor, comprehensive approach, affordance, competition, green stage

## Abstract

The use of smart devices to obtain real-time data has notably increased in the context of training. These technological tools provide data which monitor the external load and technical–tactical actions related to psychological and physical health in junior tennis players. The purpose of this paper is to monitor technical–tactical actions and physical activity during a current tennis competition in the Green stage using a Zepp Tennis Smart Sensor 2. The participants were 20 junior tennis players (under 10 years of age), with an average age of 9.46 years. The total number of strokes (n= 21,477) during 75 matches was analyzed. The study variables were the following aspects: (a) number of strokes, (b) ball impact in the sweet spot; (c) racket speed; (d) ball spin; (e) calories burned; and (f) match time. The current system of competition, based on knockout, does not meet the World Health Organization’s recommendations for daily physical activity time. Players mainly used flat forehands with a lack of variability in technical–tactical actions which did not meet the current learning opportunity criteria of comprehensive methodologies. The competition system in under-11 tennis should be adapted to the players’ characteristics by improving the variability and quantity of practice.

## 1. Introduction

The recent development of new smart devices that facilitate the access, management and transformation of information in real time is booming in elite sport contexts [1,2,3]. Companies, associations, clubs, federations, coaches, players and managers seek to acquire technological tools to improve individual and collective performance, as a means to achieve important goals [1,4,5]. The key to this growing interest lies in the synthesis of technical–tactical, physical, physiological, cinematic, biomechanical and psychological information into clear, brief and simple outputs to detect errors or deficiencies in specific variables that interact holistically with performance [1,6,7,8].

High-performance sports with a large economic and media impact, such as tennis, have focused their attention on the development and promotion of two smart devices—"smart courts and smart sensors" [9,10]. The smart court system is based on a non-invasive tool composed of a system of coordinated cameras that collect information about several factors (technical–tactical, kinematics, duration, points, etc.) in real time, with many possibilities for its use [10,11]. Furthermore, the smart sensors system provides parameters that are similar to the smart court system; however, these devices measure with less specificity, but at a lower price [10,12,13,14,15]. Easy access to smart sensors promotes their use in amateur, recreational and educational tennis contexts by coaches, educators and users [1]. Given the amount of different smart sensors on the market, it is necessary to acquire tools that fulfill the characteristics and needs of players, for the characteristics and possibilities of all players should be known, as shown in both Férnandez-García and Torres-Luque [10] and Larson and Smith’s [2] studies. 

Smart sensors are fast becoming key tools in performance analysis for decreasing the time of direct observation with optimal validity [13,16]. The symbiosis between both instruments (performance analysis and smart devices), has focused its development in high-performance sport, mainly [11] because of the desire for performance improvement and control of the training load in many sports, e.g., soccer [17,18], football [19], basketball [20], rugby [21,22], and tennis [3,13,23]. Meanwhile, the study in recreational and educational contexts is still at an early stage of development, meaning that the present and future usefulness of both tools cannot be known [24,25].

Using tools/devices that provide information about the interactions between the individual, task, environment and performance for a multidimensional improvement of the teaching–learning process in junior players agrees with the comprehensive approach [26,27]. These methodological currents offer coaches and educators a pioneering vision of the optimization of teaching–learning processes, in an ecological environment that allows for the holistic development of junior players from a creative point of view [28,29,30]. These profits are enhanced if smart sensors are used to analyze formative competition, to assess if it meets the criteria for optimization of the teaching–learning process from the comprehensive approach [25,30,31,32]. This concept is especially relevant in tennis, where the process of fluctuation and behavioral exchange offers a great dynamism in competition, which requires a meticulous analysis to enhance several factors in the formative tennis stage [3,25]. 

Coaches’ efforts using smart sensors should be focused as follows: (a) assessment of the effect of rules modification in tasks and competitions to assess if promoting the concept of affordance (quantity, variability and creativity in practice) is fulfilled [29,33]; (b) the physical–cinematic impact evaluation that sports practice entails, to adapt this to healthy parameters [33,34]; (c) the improvement of cognitive motivational variables, such as self-efficacy, satisfaction, adherence, resilience, and so on [34,35]; and (d) analysis of the individual technical–tactical pattern to correct errors or training deficiencies identified by a constraints-led approach [34,36,37]. One of the most significant current discussions is whether net height and court dimensions in the Green stage should be modified to enhance the learning opportunity under an ecological dynamics perspective [36,37]. Investigating the effect of competition in junior tennis players is a continuing concern within ecological dynamics methodology, because it is a complex process that can be simplified with the help of intelligent devices. However, there are too few studies that use smart sensors to know the impact development on the teaching–learning process in an educational context [28,34]. The aim of this research was to use a smart sensor, “Zepp Tennis 2.2.1, Zepp Labs, USA” (Zepp Tennis 2), to know the quantity and variability in physical activity and technical–tactical actions in competition (ecological context) in under-10 tennis players to analyze: (a) the number of strokes, (b) ball impact location in racket sweet spot; (c) ball speed; (d) ball spin; (e) calories burned; and (f) match time.

## 2. Materials and Methods 

### 2.1. Design and Subjects

This study was descriptive to explore the quantity and variability in formative competition [38,39]. Twenty under-10 male tennis players were selected for this study with the following anthropometric, physical–physiological and technical–tactical characteristics; (a) age of players = 9.46±0.66 years; (b) average weekly training tennis = 2.90±1.07 hours; (c) years of experience = 3.65±1.53 years; (d) weight of players = 34.80±6.59; height = 136.±7.94 cm.; (e) abdominal perimeter = 64.44±7.63; (f) “Course Navette” VO_2max_ = 20.90±4.57; (g) manual dynamometry = 14.84±3.33 kg); (h) dominant hand = 19 right hand and 1 left hand; and (i) two-handed backhand = 20. The selection criteria of the participants were accessibility and proximity (the non-randomization of the sample was for the sample’s specificity) [40]. To avoid maturity biases, a week separated both competitions [39,41]. This study respected the ethical principles established by the UNESCO Declaration on Bioethics and Human Rights. Prior to investigating, ethical clearance was obtained from the “Ethics Committee of the University of Murcia” (Spain) (ID 1925/2018). Following the Declaration of Helsinki, the players voluntarily participated in the study and their written informed consent was obtained and signed from the parents/guardians of all participants for the development of this study.

The analysis units were the strokes made in 75 matches (n = 21,477). On the first day, 38 matches were played and 37 matches were played on the second day. To control bias, a competition was designed with the following characteristics: (a) each player played four matches from one set to four games in each day (except illness or injury); (c) each player was assigned a group randomly, which comprised five players; (d) the competition system was "round robin" with a single match in each tournament; (f) rest time among matches was at least the duration of the next game (average rest = 23.60 ± 6.03 min.) to avoid cognitive, emotional and physical fatigue [36]; (g) both matches were played in the same order and schedule. The tournaments were played according to the current rules of the International Tennis Federation (Tennis 10s Green stage) for under-10 tennis players (court dimension = 23.77 × 8.23 m; net height = 0.91 m; ball= Green 25% slower than a yellow ball; age = 9–10 years; racket size = 26”; percentage of adults height = 78–82.7%; average surface=195.62 m^2^).

### 2.2. Technology Tools

The selection of Zepp Tennis 2 was made after a bibliographic review (Sport Discus, Web of Science, Google Scholar, Sponet, Scielo and Dialnet databases) searching studies that got optimal validity and reliability parameters using Zepp Tennis 2 in a real context with junior tennis players. However, the only smart sensor that could be used with all types of rackets, including junior rackets, was Zepp Tennis 2 [10]. The validation process obtained good results in ball speed and number of strokes, while for the variable kinds of strokes (e.g., slice backhand) and impact location in the sweet spot offered moderate values [13]. This device had the following specifications: (a) size (length: 1.1in/25.4mm, width: 1.1in/2.4mm, height: 0.48in/12.3mm, weight: 0.22oz/6.25g); (b) sensors (dual accelerometers, dual 3-axis gyroscopes); (c) battery (built-in rechargeable lithium ion battery, lasts up to 8 hours, 1.5 hours to fully charge); (d) connectivity Bluetooth LE; (e) location around the racket butt by “flex mount” adapter, which is transferable between rackets, besides minimizing the interference that the device can cause with the grip; and (f) use in all types of rackets, unlike other smart sensors [42].

### 2.3. Procedure

The protocol of data collection was carried out with five Zepp Tennis 2 devices as follows: (a) profile creation with sociodemographic data (links between the app and the sensor); (b) each player used the same device in all matches to avoid the data dissociation (e.g., device 1 was used for players 1, 6, 11 and 15, who played in different groups); (c) each Zepp Tennis 2 was linked to a different smartphone/APP so that there was no interference; (d) all sensors were recharged after the match so they would not discharge the battery during the competition day (between 4–5 hours). To avoid the count of previous strokes before starting the match, a designated person in charge of the smart sensors indicated when the match started. 

### 2.4. Variables and Data Notation

This study examined four variables related to the kind of strokes made and to play time. For each variable related to the kind of strokes made, there are 2 macro-categories and 12 micro-categories, as Table 1 demonstrates. The data were noted in an Excel spreadsheet; each row was a shot, while each column was a study variable (150 rows in total). Subsequently, an exploratory data analysis was done for performing the initial investigation, as well as identifying errors and anomalies in the data with summary statistics [43]. The database was transformed for the analysis of variance (each group is a kind of stroke) and a factor was included (categorical variable) for analyzing the number of strokes, the percentage of ball impact in the sweet spot, ball speed and ball spin.

### 2.5. Statistical Analysis

Data analyses were divided into five phases: (a) descriptive analysis; (b) analysis of variance; (c) non-clinical magnitude-based inference; (d) effect size; and (e) linear regression analysis. Descriptive analysis was performed using mean (M) and standard deviation (SD). The comparisons between variables were assessed using one-way ANOVA [44]. To assess the homoscedasticity, the Levene test for equal variances was checked. It was appreciated that the data did not meet this criterion, not assuming equal variances, and the post-hoc Games–Howell was adopted for better controlling the error rate in different post-hoc analyses. For the post-hoc analysis, the alpha level for statistical significance was set at p < 0.05 [44]. Non-clinical magnitude-based inference was applied to the post-hoc comparison with standardized mean differences and their 90% confidence intervals (CI) were reported [45,46]. Prior to the comparison, all variables were log-transformed to reduce the bias of non-uniformity errors and because the range of some values plotted was greater than ~×1.25 [47]. The smallest important magnitude was calculated as 0.2 times the between-matches standard deviation, as reported by Gimenez, Leicht and Gomez [48]. The effect that was reported was unclear in showing if the CI overlapped positive and negative values simultaneously [47]. The magnitudes of clear effects were described in the following qualitatively scale: 25%–75%, possibly; 75%–95%, likely; 95%–99%, very likely; > 99%, most likely [49]. The effect size in the post-hoc comparison was interpreted using the following criteria: ≤0.2 = trivial; 0.2-0.6 = small; 0.6-1.2 = moderate; 1.2-2.0 = large; 2.0-4.0 = very large; >4.0 = extremely large [47]. Finally, to observe the strength of the association between the calories burned and the match time, a linear regression was reckoned. Statistical analyses were performed using the IBM SPSS Statistics 23.0 statistical package (IBM Corp., Armonk, NY, United States) and Excel spreadsheets “compare 2 or 3 groups mean” designed by Batterham and Hopkins [45]. 

## 3. Results

The descriptive values (mean ± standard deviation) and analysis of variance (one-way ANOVA “post hoc”) regarding the kind of strokes made per match divide in basic strokes and groundstrokes are shown in Table 2. The pairwise comparisons between six types of basic strokes and all groundstrokes, based on the estimation of true differences and standardized Cohen’s d (effect size ± confidence interval), can be observed in Figure 1. The linear regression between calories burned and match time are presented in Figure 2, which observes their relationship. 

The analysis of variance in Table 2 indicates statistically significant differences in the following variables: (a) number of strokes in basic strokes (p = 0.000; F = 59.323; df1 = 5; df2 = 889; η2 = 0.250) and groundstrokes (p = 0.000; F = 75.443; df1 = 5; df2 = 889; η2 = 0.298); (b) ball impact in the sweet spot in basic strokes (p = 0.000; F = 10.104; df1 = 5; df2 = 742; η2 = 0.064), and groundstrokes” (p = 0.000; F = 24.681; df1 = 5; df2 = 840; η2 = 0.128); (c) ball speed in basic strokes (p = 0.000; F = 371.258; df1 = 5; df2 = 742; η2 = 0.741) and groundstroke (p = 0.000; F = 32.935; df1 = 5; df2 = 840; η2 = 0.164); and (d) ball spin in basic strokes (p = 0.000; F = 51.394; df1 = 5; df2 = 742; η2 = 0.257) and groundstrokes classification (p = 0.000; F = 296.338; df1 = 5; df2 = 840; η2 = 0.638).

Results in Table 2 and Figure 1 show the highest values of hitting with the dominant hand. Players used three times more forehands and forehand volleys than backhands and backhand volleys. When comparing the number of flat strokes (55.09%) with the total topspin and slice strokes (16.93-27.96% respectively), it can be seen that almost half of the shots were flat.

Additionally, the backhand showed a higher percentage of sweet spot impact than forehand impact. The mean percentage of serves and smashes (strokes made over the shoulder) was the lowest percentage in the variable “ball impact in the sweet spot”, not achieving 70%. When observing the topspin forehand impact location, it can be seen that this had the lowest value of all groundstrokes (over 40% did not impact in the sweet spot).

The ball speed can be expressed in a decreasing order of speed, as follows: (a) serve and smashes (values near to 100 km/h); (b) flat groundstrokes (approximately 80 km/h); (c) slice and topspin groundstrokes (values between 70 and 75 km/h); and (d) volley strokes (speed below 70 km/h).

The results disclosed that the backhand gets more than 100 rpm on average than the forehand. Specifically, the highest value above 2000 rpm was the slice backhand. In relation to the quantification of rpm in groundstrokes, the lowest ball spin values corresponded to the flat strokes at around 1000 rpm.

The descriptive analyses of “match time” and “calories burned” had the following values: match time: *M*= 21.80; min. = 61; max. = 95; SD = 12.30/calories burned: *M* = 179.00; min. = 85; max. = 495; SD = 96.10. In addition, the sum of the mean values of four matches played in one day was 87.20 min and 716 kilocalories. The relationship between both variables can be seen in Figure 1, linear regression showed that 73% of the burned calories variability was associated with the number of minutes played per match.

## 4. Discussion

The aim of this research was to use a smart sensor “Zepp Tennis 2” to know the quantity and variability in physical activity and technical–tactical actions in competition (ecological context) in under-10 tennis players to analyze: (a) the number of strokes, (b) ball impact location in the sweet spot; (c) ball speed; (d) ball spin; (e) calories burned; and (f) match time. 

As mentioned in the literature and in this research, the players showed a clear predisposition to using forehand strokes matching their dominant side, because they felt secure and effective in their comfort zone [50,51]. However, these results differ from the concept of variability to enhance learning opportunities in the formative stage. These results match those observed previously, stating the need to improve backhand ratios, because players who use their backhand when playing are more likely to perform successful hitting sequences to win points [52]. Continuing with variability, there was an inclination towards the use of a flat forehand and backhand over other groundstrokes. However, these results differ from some published studies on the Green stage, which show the predominant use of topspin [36,37]. It is probable that the target of young players was to hit with an exaggerated topspin to force their opponents to hit the ball over their shoulders, outside their comfort area, taking advantage of the high bouncing of the ball and court dimensions. There are several explanations for these results, namely: (a) the players’ characteristics of development in this study differed from the studies reviewed; and (b) a lack of technical–tactical intelligence and transference to a real game context. One finding related to variability was that the best mean value obtained with the “ball spin” variable matched the backhand slice, with values above 2000 rpm. Although the backhand slice got a high spin, the players only used it 5.60% of the time. The under-10 tennis players were not aware of this fact limiting their playing ability; consequently, there was little variability at this early training stage.

Moreover, the results of the flat groundstrokes showed a higher speed than other types of groundstrokes; there likely exists a connection between swing speed and the number of flat groundstrokes. These results can be explained, in part, because young tennis players use flat groundstrokes to win points through this tactic. These findings did not show a variability and creativity behavior, as ecological-dynamic methodologies suggested [53,54]. However, these findings suggested that the players take on more offensive behaviors, an aspect that these methodologies seek to improve by positively influencing the satisfaction and sport adherence [30,54].

One issue that emerged from these findings is the greater number of volleys than serves per match. The researchers, from anecdotal evidence, observed that Zepp Tennis 2 confused serves with forehand volleys, because the children unconsciously modified their pattern of serve, skipping the preparatory phase for not serving out of the service box, decreasing the serve speed. These results are more likely to occur due to the lack of courts and nets adapted to the children’s characteristics, as promoted by the “scaling equipment perspective” [33,37,55].

In relation to wellbeing, the World Health Organization recommends moderate or vigorous physical activity of at least 60 minutes a day for children and youths, however, over 60% of matches do not even last 25 minutes [56]. The current competition system, based on knockout, does not meet the daily minimum values of physical activity established by the World Health Organization. Therefore, a round robin competition system seems more effective for achieving this well-being criterion.

These findings, while preliminary, suggest that the use, mainly, of one kind of stroke or dominant side does not encourage the variability by playing associated with creative problem-solving [57]. One possible explanation for these results may be the lack of training in global and integral situations to develop different skills and behaviors in problem-solving. [58]. Another possible factor that does not allow players to perform creative and varied behaviors may be a lack of adaptation of the rules, playing spaces and sports equipment to meet the needs of players in Green competitions [37,58]. According to Grambow et al. [59], is important extrapolate these findings, working underutilized strokes into the training process at the Green stage, supporting variability in practice in the long and short term.

The main weakness of this study was the paucity of validation studies with smart sensors that are compatible with all types of junior rackets (the sample’s characteristics) in the early tennis stage [3,5,13,42]. The only research that used a smart sensor that could be mounted on junior rackets was carried out on high-performance players by Keaney and Reid [13]. The study validation showed a good accuracy regarding the count of backhand and forehand strokes besides the ball speed variable, while moderate imprecision detected the kind of stroke in relation to the kind of spin (e.g., slice backhand). These limitations were also found in other types of studies with smart sensors [5,60], while, in others, the number of strokes executed minimized this effect due to the statistical frequency analysis [3]. 

This research has thrown up many questions that are in need of further investigation. Future work needs to be done with different samples and contexts to know the effect of competition in real time using smart devices. This type of study using smart sensors was, in full development, intended to look for deficiencies in the teaching–learning processes, despite their limitations [5,42,60]. Finally, deeper research should be carried out with coaches, educators, engineers and universities to adjust these technological tools to accurately identify smashes, serves and forehand volleys in junior tennis players [1,12]. 

## 5. Conclusions

In summary, under-10 tennis players in the Green competition stage grounded their play in forehands, especially flat forehands. It is possible that they gained an advantage from assuming an offensive behavior by increasing the speed of the ball, even though it involves a greater risk. Although using flat forehands is desirable for offensive play, it is not advisable to overuse it from the point of view of hitting variability. This lack of play variability limits learning by not providing different experiences that encourage creativity processes in problem-solving at an early stage [58]. In addition, the round robin system enhanced the time of physical activity to achieve healthy values, however, the current competition, based on knockout, does not ensure that all participants reach a practice time over 60 minutes. The present study confirms previous findings and contributes additional evidence that suggests redesigning Green competitions to improve the teaching–learning process for under-10 tennis players [37,58].

To conclude, Zepp Tennis 2 allows coaches to access information in real time and without effort to adapt teaching–learning processes both in competition and training, improving the physical and technical–tactical levels of the players at the Green stage at a low cost [3,9].

## 6. Practical Applications

The most obvious finding to emerge from this study is that coaches and federations should redesign Green competitions, reducing the net height and court dimensions, encouraging backhand scoring [51] and setting a round robin format. In agreement with Bayer et al. [37], the first proposal would reduce net height and court dimensions (net height = 0.85 m and court dimensions = 20.77 × 7.19 m) for the following reasons: (a) volleys would be easier, because the net is closer and smaller, also the court would be less wide so that it covers the whole net; (b) the shorter court would force the players to hit topspin groundstrokes so that the ball drops to the ground quicker (Magnus effect) and players would be encouraged to use them. Grounded in the proposal conducted by Gimenez-Egido et al. [58], was an interest in developing a competition with the following characteristics: (a) a round robin format; (b) each match is played in a different scaled court (1^st^ match = 18.007 × 7.19 m; 2^nd^ match = 18.007 × 8.23 m; 3^rd^ match = 20.777 × 7.19 m; 4^th^ match = 20.777 × 8.23 m); (c) net height=0.80 m; and (d) points won with backhand are worth two points. This competition would create different experiences and opportunities to exploit players’ learning to the fullest under ecological–dynamics methodologies. 

Coaches, based on these competition parameters set out above, should design training tasks, manipulating the constraints to enhance the opportunities between the player and the task to improve this formative stage [58].

## Figures and Tables

**Figure 1 ijerph-17-01068-f001:**
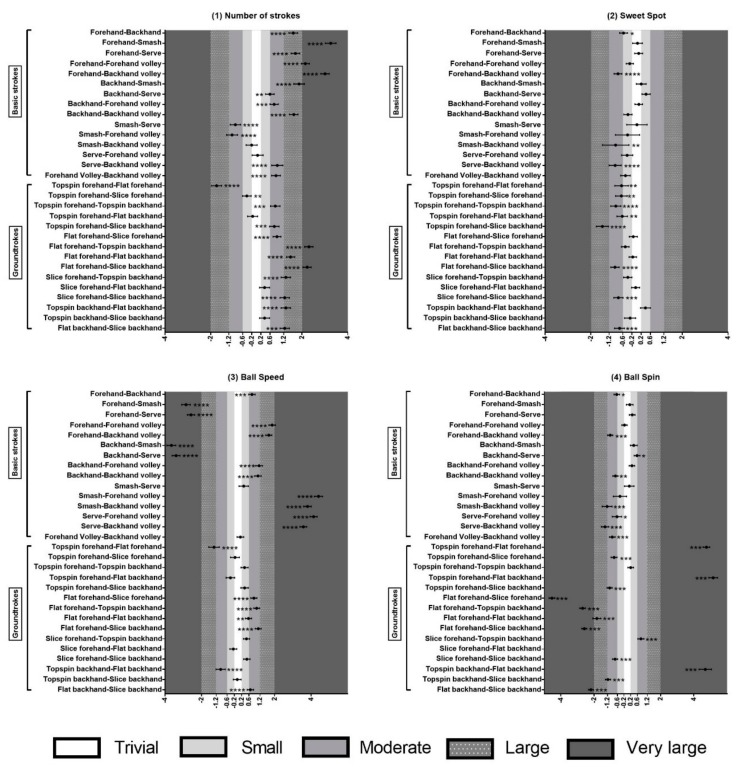
Standardized effect size in post-hoc comparison between basic strokes and between groundstrokes in: 1) number of strokes; 2) sweet spot; 3) ball speed; 4) ball spin during Green stage competition. The asterisks show the likelihood of the magnitude of the true effect in the following scale: ∗possibly, ∗∗likely; ∗∗∗very likely; ∗∗∗∗most likely. The white area between –0.2 and 0.2 reveals trivial differences; as the area darkens, the differences are larger. The standardized value direction depends on the relationship between strokes.

**Figure 2 ijerph-17-01068-f002:**
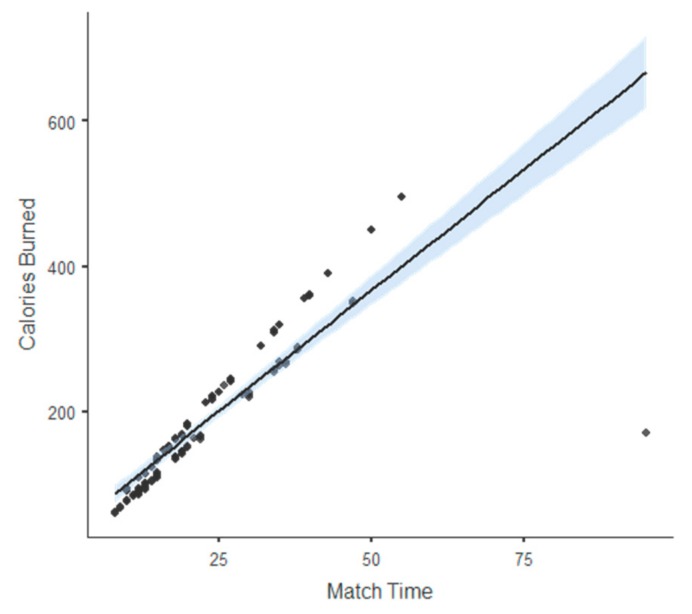
Linear regression between calories burned (unit of measurement: “kilocalories”) and match time (unit of measurement: “minutes”).

**Table 1 ijerph-17-01068-t001:** Description of variables (kind of stroke, percentage of impact location in the sweet spot, ball speed, ball spin, calories burned and match time) and description of categories of study.

Variables (Description)	Categories	Sub-Categories	Sub-Categories (Description)
1. Number of strokes (total number of strokes per match) 2. Percentage of ball impact in the sweet spot (percentage of hits located in the sweet spot) 3. Ball Speed (average speed measures in kilometers per hour) 4. Ball Spin (spin measures in revolutions per minute)	Basic strokes	Forehand	Groundstroke on the dominant side per match
		Backhand	Groundstroke on the non-dominant side per match
		Smash	Stroke above the hitters’ head with a serve-like motion
		Service	Stroke to start the point. Normally is served overhead
		Forehand Volley	Shot on the dominant-side in which the ball is hit before it bounces on the ground (not smash)
		Backhand Volley	Shot on the not-dominant-side in which the ball is struck before it bounces on the ground (not smash)
	Groundstrokes	Topspin Forehand	Topspin shot on the dominant side
		Flat Forehand	Flat shot on the dominant side
		Slice Forehand	Slice shot on the dominant side
		Topspin Backhand	Topspin shot on the non-dominant side
		Flat Backhand	Flat shot on the non-dominant side
		Slice Backhand	Slice shot on the non-dominant side
5. Calories burned	-	-	Kilocalories burned per match
6. Match time	-	-	Play time in minutes

**Table 2 ijerph-17-01068-t002:** Descriptive values (mean and standard deviation) and analysis of variance (ANOVA one-way) on number of strokes, percentage of hits in sweet spot, ball speed and ball spin.

	NUMBER OF STROKES		SWEET SPOT		BALL SPEED		BALL SPIN	
Var	Mean ± SD	post-hoc	Mean ± SD	post-hoc	Mean ± SD	post-hoc	Mean ± SD	post-hoc
Fh	88.70±61.95	**Fh**>(Bh, Sm, Se, FhV, BhV)	68.00±10.20	Bh >**Fh**>(FhV, BhV)	79.50±6.16	Bh**<Fh<**(Sm, Se, FhV, BhV)	1224.00±152.00	(Bh, BhV)>**Fh**
Bh	24.10±16.83	Fh>**Bh**<(Sm, FhV, BhV)	76.20±15.40	**Bh**>(Fh, Se, Sm)	74.40±6.35	**Bh**<(Fh, Sm, Se, FhV, BhV)	1367.00±234.00	BhV>**Bh**
Sm	2.27±1.59	(Fh, Bh, FhV, BhV)>**Sm**	64.30±35.00	(Bh, BhV, FhV)>**Sm**	99.20±14.50	**Sm**>(Fh, Bh, FhV, BhV)	1223.00±420.00	BhV>**Sm**
Ser	16.79±7.12	Fh>**Se**	67.30±23.10	(Bh, FhV, BhV)>**Se**	98.20±7.96	**Se**>(Fh, Bh, FhV, BhV)	1158.00±370.00	Bh, FhV, BhV>**Se**
FhV	12.67±8.85	(Fh,Bh)>**FhV**> (Sm, BhV)	74.30±21.40	**FhV**>(Fh, Sm, Se)	66.90±6.22	(Fh, Bh, Sm, Se)>**FhV**	1297.00±269.00	BhV>**FhV**
BhV	5.33±3.72	(Fh, Bh, FhV)>**BhV**>Sm	81.60±25.10	**BhV**>(Fh, Sm, Se)	66.10±7.19	(Fh, Bh, Sm, Se)>**BhV**	1818.00±662.00	(Fh, Bh, Sm, Se, FhV)>**BhV**
TsFh	13.70±9.57	(FFh,SFh)>**TsFh**> (TsBh, SBh)	58.50±24.20	(FFh, SFh, TsBh, FBh, SBh)>TsFh	74.40±9.97	FFh>**TsFh**>(TsBh, SBh)	1494.00±166.00	(FFh, FBh<**TsFh**< SFh, SBh)
FFh	49.30±34.43	**FFh**>(TsFh, SFh, TsBh, FBh, SBh)	68.50±11.50	TsFh<**FFh**<(TsBh, SBh)	81.60±6.05	**FFh**>(TsFh, SFh, TsBh, FBh, SBh)	931.00±91.50	(TsFh, SFh, TsBh, FBh, SBh)>**FFh**
SFh	25.60±17.88	(TsFh, FFh)> **SFh**>(TsBh, FBh, SBh)	71.80±16.30	TsFh<**SFh**>SBh	75.30±7.07	(FFh>**SFh**>TsBh, SBh)	1656.00±217.00	SBh>**SFh**>(TsFh, FFh, TsBh, FBh)
TsBh	5.73±4.00	(TsFh, FFh, TsBh,FBh)>**TsBh**	78.50±26.70	**TsBh**>(TsFh, FFh)	70.70±10.50	(TsFh, FFh, SFh, FBh)>**TsBh**	1458.00±265.00	(SFh,SBh)>**TsBh**>(FFh, FBh)
FBh	13.90±9.71	(FFh,SFh)>**FBh**>(TsBh, SBh)	73.20±18.40	TsFh<**FBh**<SBh	77.20±7.62	FFh>**FBh**>(TsBh, SBh)	1050.00±71.60	(TsFh, SFh, TsBh, SBh>FBh)>**FFh**
SBh	6.48±4.53	(TSF, FFh, SFh, FBh)>**SBh**	83.70±22.10	**SBh**>(TsFh, FFh, SFh, FBh)(	71.00±8.98	(TsFh, FFh, SFh, FBh)>**SBh**	2163.00±656.00	**SBh**>(TsFh, FFh, SFh, TsBh, FBh)

Var = Variables; Basic strokes (Fh = Forehand; Bh = Backhand; Sm = Smash; Se = Serve; FhV = Forehand Volley; BhV = Backhand Volley). Groundstrokes (TsFh = Topspin Forehand; FFh = Flat Forehand; SFh = Slice Forehand; TsBh = Topspin Backhand; FBh = Flat Backhand; SBh = Slice Backhand.

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
