# Peer review of "Using Smart Sensors to Monitor Physical Activity and Technical–Tactical Actions in Junior Tennis Players"

_ijerph, 2020, doi:10.3390/ijerph17031068_

Round 1

Reviewer 1 Report

In my opinion the paper is well written and is ready for publication.

The only weak issue is tab.1. I am not able to read it with proper care due to it's size is too small. I strongly suggest to creat a bigger one.

Please concern also citing the following paper:

"Serve efficience developments at Wimbledon between 2002 and 2015: a longitudinal approche to impact tomorrow's tennis practice.

Human Movement 2020 vol. 21 (1), 65-72 DOI: https://doi.org/10.5114/hm.2020.88155

Author Response

Rebuttal letter

Reviewer: In my opinion the paper is well written and is ready for publication.

Reviewer: The only weak issue is tab.1. I am not able to read it with proper care due to it's size is too small. I strongly suggest to creat a bigger one.

Authors: Thank you for your feedback. This table has been modified.

Reviewer: Please concern also citing the following paper:

"Serve efficience developments at Wimbledon between 2002 and 2015: a longitudinal approche to impact tomorrow's tennis practice.

Human Movement 2020 vol. 21 (1), 65-72 DOI: https://doi.org/10.5114/hm.2020.88155

Authors: Thank you very much for your contribution. This article has been included in the manuscript.

Reviewer 2 Report

Discussing the results, the authors use very unreadable comparisons to the work of other researchers. Comparing one issue to 22 or 31 references may indicate that the authors conducted studies that have already been conducted many times and there is no element of originality in them (verses 261: 43-65; verses 262 and 283: 28-59). At the same time, among the listed publications there are those that are not entirely related to the problem being examined (e.g. item 37 concerns handball regulations; item 38 relates to physical activity and childhood obesity, items 39 and 40 apply to football players, etc.). Verifications of the cited references should be carried out.

The conclusions formulated in this way are hardly readable. The conclusions should be divided: in the first group there should be conclusions from the conducted research, and in the second group - the conclusions for practical application.

Author Response

Reviewer: Discussing the results, the authors use very unreadable comparisons to the work of other researchers. Comparing one issue to 22 or 31 references may indicate that the authors conducted studies that have already been conducted many times and there is no element of originality in them (verses 261: 43-65; verses 262 and 283: 28-59). At the same time, among the listed publications there are those that are not entirely related to the problem being examined (e.g. item 37 concerns handball regulations; item 38 relates to physical activity and childhood obesity, items 39 and 40 apply to football players, etc.). Verifications of the cited references should be carried out.

Authors: Thank you very much for your contribution. Following your recommendations, the discussion section has been reworded to improve its readability. In addition, literature exclusively related to tennis and ecological-dynamic perspective has been used to rewrite this section.

Reviewer:  The conclusions formulated in this way are hardly readable. The conclusions should be divided: in the first group there should be conclusions from the conducted research, and in the second group - the conclusions for practical application.

Authors: We thank you again for your contribution. The conclusions section has been structured according to your guidelines.

Reviewer 3 Report

Comments

The purpose of this article was to monitor technical-tactical actions and physical activity during a current tennis competition in Green stage using Zepp Tennis Smart Sensor 2 by focusing on number of strokes, ball impact location on racket sweet spot, ball speed, ball spin, calories burned and match time. This aim is relevant and appropriate for this journal. However, I have several comments:

Introduction

- It is not clear why using very young tennis players (under 10 years old). Why not older subjects with more experience and skills? Authors should justify their choice of this population to test their device.

- The relevance of each variable (number of strokes, ball impact location on racket sweet spot, ball speed, ball spin, calories burned and match time) was not explained. Which variables is considered as measurement of tactical performance in tennis?

- It is unclear what authors considered as “weaknesses » in the aim of the study. Authors need to be more explicit to explain what does that mean here. It complicates reading of the article.

Methods

How calories burned are measured with this device? How can authors explain the relevance of this measurement with this device? How the ball speed can be measured with Zepp Tennis Smart Sensor 2?

Results

- Be careful: Results show in Table 1 and Figure 1 show large and substantial differences among different 185 kinds of strokes. Strokes. (Repetition to remove, typing error…).

- Results are hard to read. Sentences need to be more concise to be clearer. What we have to understand? Please rewrite sentences.

- Figure 1 is not clear. “This figure has multiple panels and they are listed as: …». Remove this sentence and write what we have to know to read this figure please.

Discussion

This part is confusing for me. I’m not sure to understand what the authors want to highlight. The notion of cognitive-constructivist and ecological-dynamic only appear in the conclusion section without a clear definition. How the discussion answers to the question of the article?

It is really hard to read.

Author Response

Rebuttal letter

Introduction

Reviewer: - It is not clear why using very young tennis players (under 10 years old). Why not older subjects with more experience and skills? Authors should justify their choice of this population to test their device.

Authors: Thank you for your feedback. We have included the justification for the choice of the sample in this section.

From line 71 to 73 “One of the most significant current discussion is whether net height and court dimensions in green stage should be modified to enhance learning opportunity under ecological dynamics perspective”.

Reviewer: - The relevance of each variable (number of strokes, ball impact location on racket sweet spot, ball speed, ball spin, calories burned and match time) was not explained. Which variables is considered as measurement of tactical performance in tennis?

Authors: - Thank you for your comment. We understand your perspective but the only variables checked in other studies with Zepp Tennis 2 are the ones used in this study.

Reviewer: - It is unclear what authors considered as “weaknesses » in the aim of the study. Authors need to be more explicit to explain what does that mean here. It complicates reading of the article.

Authors: Thank you for your advice. The term weakness has been replaced throughout the text (see line 2).

Methods

Reviewer: How calories burned are measured with this device? How can authors explain the relevance of this measurement with this device?

Authors: The measurement of calories burned is found in the user guide Zepp Tennis 2. The calculate of calories burned base on the following formula: (MET x body weight x 3,5) /200 x match time.

Reviewer: How the ball speed can be measured with Zepp Tennis Smart Sensor 2?

Authors: In previous studies, ball speed measure by Zepp Tennis 2 both groundstrokes and serve has a correlation coefficient 0.97 with VICON system (500-Hz 12 cameras).

Results

Reviewer: Be careful: Results show in Table 1 and Figure 1 show large and substantial differences A

Authors: Exploration analysis was done previously to perform the statistical analysis to control data quality.

From line 143 to 145: “Subsequently, an exploratory data analysis was done for performing initial investigation, as well as identifying errors and anomalies data with summary statistics”.

Reviewer: - Results are hard to read. Sentences need to be more concise to be clearer. What we have to understand? Please rewrite sentences.

Authors: We have rewritten this section.

Reviewer 3: - Figure 1 is not clear. “This figure has multiple panels and they are listed as: …». Remove this sentence and write what we have to know to read this figure please.

Authors: Thank you for your advice. We have included how to read Figure 1.

Discussion

Reviewer 3: This part is confusing for me. I’m not sure to understand what the authors want to highlight. The notion of cognitive-constructivist and ecological-dynamic only appear in the conclusion section without a clear definition. How the discussion answers to the question of the article?

Reviewer: It is really hard to read.

Authors: Thank you very much for your contribution. Following your recommendations, the discussion section has been reworded to answer the question of the article. In addition, we have included in the introduction section information about ecological-dynamic perspective linked with the discussion section.

Round 2

Reviewer 2 Report

In response to the review, there is a sentence about adapting the conclusions to the reviewer's comments, yet this has still not been done in the article.

There are still references to many publications in one sentence, e.g. inee 64 [3-25].

Does the entry [29-37-40] in line 67 refer to three publications? Is it a range of 29-40 items from the list of publications?

The same situation requires clarification regarding line: 69 [35-39-42], 77 [28-35], 253 [34-45-64], 265 [45-67], 275 [5-69 ], 280 [5-50-69] and 283 [1-12].

Author Response

Reviewer:

In response to the review, there is a sentence about adapting the conclusions to the reviewer's comments, yet this has still not been done in the article.

Authors:

Please excuse us if we didn't know how to reflect this in the manuscript. We have modified the manuscript including a new section called Practical Application besides us rewriting the conclusions.

Reviewer:

There are still references to many publications in one sentence, e.g. inee 64 [3-25].

Does the entry [29-37-40] in line 67 refer to three publications? Is it a range of 29-40 items from the list of publications?

The same situation requires clarification regarding line: 69 [35-39-42], 77 [28-35], 253 [34-45-64], 265 [45-67], 275 [5-69 ], 280 [5-50-69] and 283 [1-12].

Authors:

Thank you for your feedback. We have corrected these errors changing “-“ to “,”.

Reviewer 3 Report

Authors have answered to my concerns.

thanks

Author Response

Rviewer

Authors have answered to my concerns.

thanks

Authors

Thank you very much for your review. The manuscript has improved greatly.